# *Pistacia vera* Extract Potentiates the Effect of Melatonin on Human Melatonin MT_1_ and MT_2_ Receptors with Functional Selectivity

**DOI:** 10.3390/pharmaceutics15071845

**Published:** 2023-06-28

**Authors:** Nedjma Labani, Florence Gbahou, Marc Noblet, Bernard Masri, Olivier Broussaud, Jianfeng Liu, Ralf Jockers

**Affiliations:** 1Cellular Signaling Laboratory, International Research Center for Sensory Biology and Technology of MOST, Key Laboratory of Molecular Biophysics of Ministry of Education, School of Life Science and Technology, Huazhong University of Science and Technology, Wuhan 430074, China; nedjma.labani@gmail.com (N.L.); jfliu@mail.hust.edu.cn (J.L.); 2Institut Cochin, CNRS, INSERM, University of Paris, F-75014 Paris, France; florence.gbahou@inserm.fr (F.G.); bernard.masri@inserm.fr (B.M.); 3Science Hub, Sanofi Consumer Healthcare, F-75017 Paris, France; marc.noblet@sanofi.com (M.N.); olivier.broussaud@sanofi.com (O.B.)

**Keywords:** plant extracts, GPCR, signalling, melatonin, insomnia, depression

## Abstract

Melatonin is a tryptophan derivative synthesized in plants and animals. In humans, melatonin acts on melatonin MT_1_ and MT_2_ receptors belonging to the G protein-coupled receptor (GPCR) family. Synthetic melatonin receptor agonists are prescribed for insomnia and depressive and circadian-related disorders. Here, we tested 25 commercial plant extracts, reported to have beneficial properties in sleep disorders and anxiety, using cellular assays (2─[^125^I]iodomelatonin binding, cAMP inhibition, ERK1/2 activation and β-arrestin2 recruitment) in mock-transfected and HEK293 cells expressing MT_1_ or MT_2_. Various melatonin receptor-dependent and -independent effects were observed. Extract 18 (**Ex18**) from *Pistacia vera* dried fruits stood out with very potent effects in melatonin receptor expressing cells. The high content of endogenous melatonin in **Ex18** (5.28 ± 0.46 mg/g extract) is consistent with this observation. **Ex18** contains an additional active principle that potentiates the effect of melatonin on G_i_ protein-dependent pathways but not on β-arrestin2 recruitment. Further active principles potentiating exogenous melatonin were detected in several extracts. In conclusion, we identified plant extracts with various effects in GPCR-based binding and signalling assays and identified high melatonin levels and a melatonin-potentiating activity in *Pistacia vera* dried fruit extracts that might be of therapeutic potential.

## 1. Introduction

In traditional medicine practiced around the world, plants are the primary agents used for treating illness. In many countries, plant extracts are sold as over-the-counter formulations for self-medication, or are prescribed by physicians as illustrated by the 294 listed traditional herbal Kampo medicines in Japan [1]. Botanically “defined mixtures” are now recognized as drug entities by the Food and Drug Administration (FDA) [2] and among all FDA-approved new molecular entities from natural sources, 25% are derived from plants [3]. Many medicinal plants act on functions of the central nervous system (CNS) which are mainly under the control of ion channels and GPCRs. Not surprisingly, earlier studies in the nineties focused on the identification of CNS-active principles by screening plant extracts against a series of GPCR radioligand binding assays [4]. In several cases, the most active principle has been identified [1,2]. Plant-derived molecules may span the full range of GPCR ligands including competitive ligands binding to the orthosteric ligand binding site or molecules with functional selectivity [5]. An increasing number of studies report molecules with allosteric action mechanisms. The flavonoid quercetin has been proposed to bind to an allosteric binding site of visual rhodopsin [6]. T20K, a plant-derived disulfide-rich peptide comprising a cyclic cystine knot, was shown to act on the kappa opioid receptor according to a mixed orthosteric/allosteric binding mode [7]. 

Among the different GPCRs, MT_1_ and MT_2_ receptors are targeted by the neurohormone N-acety-5-methoxyltryptamine or melatonin (MLT), an important regulator of circadian and seasonal rhythmicity, retinal and immune functions, glucose homeostasis and sleep [8,9,10,11]. Several plant extracts have been tested on melatonin receptors [12]. Valerian and hops extracts, described as sleep aid, were able to compete with the melatonin receptor-specific 2─[^125^I]iodomelatonin (2─[^125^I]─MLT) radioligand with apparent IC_50_ values of 100–450 µg/mL [13]. Another study provides correlative evidence showing that a *Piper betle* L. leaf extract, at 50 µg/mL, has similar beneficial effects as MLT in an early stress model in zebrafish [14]. Isolated plant-derived molecules were also tested. Oxyprenylated ferulic acid and several coumarin derivatives showed interesting properties in the 2─[^125^I]─MLT competition binding assay with K_i_ values in the high nM to low µM range for MT_1_ that, however, remained partial up to concentrations of 100 µM, which is suggestive of a more complex, allosteric, binding mode [15]. Another series of articles report the isolation of plant-derived compounds from various species used in Chinese traditional medicine with modest EC_50_ values ranging widely from 10 µM to 1 mM for MT_1_ and MT_2_ [16,17,18,19,20,21]. Functional assays were performed with an engineered calcium-reporter cell line expressing MT_1_ but no counter-screen was performed on control cell lines devoid of MT_1_ expression. Similar screening studies with other GPCRs indicated that such controls are mandatory to demonstrate specificity for the target receptor [22]. 

Here, we selected 25 commercially available plant extracts to identify those containing melatonin-like activities. Based on the known modulatory properties of melatonin receptor ligands on sleep and depression, extracts from plants reported to have beneficial properties in reducing sleep disorders and anxiety were selected (for more details on the choice of plants, see the Methods Section) [23,24]. A similar rationale has been successfully applied previously to the identification of plant-derived molecules acting on adenosine receptors [25]. First, the 25 plant extracts were tested in the 2─[^125^I]─MLT competition binding assay and based on their K_i_ values 10 extracts were selected for further characterization in three functional assays (inhibition of cAMP production, ERK1/2 activation, and β-arrestin2 recruitment) in mock-transfected HEK293 cells and cells expressing either MT_1_ or MT_2_. All extracts were also tested for the presence of an allosteric activity that potentiates the effect of exogenously added melatonin.

## 2. Materials and Methods

### 2.1. Plant Extracts

Commercially available extracts (Table 1) were selected based on the following references reporting beneficial properties of plants in reducing sleep disorders and anxiety: for **Ex1**, and **Ex6** (*Melissa officinalis*); **Ex2**, and **Ex5** (*Passiflora incarnata*); **Ex4** (*Crocus sativus*); **Ex3**, and **Ex7** (*Eschscholzia californica*), **Ex9** (*Hypericum perforatum*), **Ex12** (*Humulus lupulus*), **Ex13** (*Withania somnifera*), **Ex19** (*Rhodiola rosea*); **Ex24**, and **Ex25** (*Valeriana offinalis*) see [23,24]; for **Ex12**, **Ex24**, and **Ex25** see [25]; for **Ex14** (*Withania somnifera*) see [26]; for **Ex18** (*Pistacia vera*) see [27]; and for **Ex23** (*Verbena officinalis*) see [28].

### 2.2. Test of Solubility

As the primary screen of all extracts was performed with the radioligand binding assay, we tried to first solubilize the lyophilized plant extracts in the radioligand binding TEM buffer (75 mM Tris, 5 mM EDTA 12.5 mM MgCl_2_, and pH = 7.5) [29]. Most of the extracts were soluble in this buffer at 40 mg of each extract powder per 250 µL of TEM, except **Ex18**, for which the ratio was 1 mg of extract powder solubilized in 2 mL of solvent. The remaining extracts were solubilized in DMSO (Sigma Aldrich, Saint-Louis, MO, USA) and a sonication step was followed whenever necessary.

### 2.3. Cell Culture of HEK293 Cells

HEK293 cells were grown in complete DMEM medium, supplemented with 10% (V.V^−1^) FBS, 4.5 g.L^−1^ glucose, 100 U.mL^−1^ penicillin and 0.1 mg.mL^−1^ streptomycin and incubated at 37 °C (95% O_2_, and 5% CO_2_). HEK293 cell lines stably expressing human MT_1_ or MT_2_ were described previously [30]. Transient transfection of cells was performed with JetPei reagent according to the supplier’s instructions (Polyplus Transfection, NY, USA).

### 2.4. 2─[^125^I]─MLT Radioligand Binding Assay 

Membranes from HEK293 cells stably expressing human MT_1_ or MT_2_ were prepared as previously described [29] and use in radioligand binding assay. Briefly, the competition binding assay was performed by simultaneous incubation of [^125^I]-MLT (100 pM) and increasing concentration of MLT or plant extracts. The assay was carried out in 120 min at 37 °C followed by a rapid filtration through GF/C glass fibre filters (Whatman, Clifton, NJ, USA). Filter-retained radioactivity was determined with a γ-counter LB2111 (Berthold Technologies, Bad Wildbad, Germany). The efficiency of the extract K_i_ is expressed by the IC_50_ of the extract normalized to the concentration of the radioligand by using the Cheng–Prussof formula: K_i_ = IC_50_/[1 + (L/K_d_)], where, L represents the ligand concentration and K_d_ the dissociation constant. K_d_ values were 152 and 367 pM for MT_1_ and MT_2_ receptors respectively.

### 2.5. Accumulative cAMP Assay

The cAMP assay was performed as previously described [31]. Briefly, HEK293 cells transiently transfected with empty vector (mock) or stably expressing human MT_1_ or MT_2_ were dispensed into a 384-well plate (4000 cells per well) and stimulated with 1µM forskolin alone or in the presence of increasing concentrations of MLT or extracts, incubated for 30 min at room temperature in stimulation buffer supplemented with 1 mM IBMX (Sigma-Aldrich, St Quentin, France). Cells were then lysed and, cAMP content was determined following the supplier’s instructions (PerkinElmer CisBio cAMP-G_i_ kit, Codolet, France). The plate was read using the Infinite F500 Tecan microplate reader. Data were fitted via non-linear regression to determine EC_50_ and E_max_ values using GraphPad Prism software (La Jolla, CA, USA).

### 2.6. ERK1/2 Activation Assay

A quantitative analysis of ERK1/2 activation in non-transfected HEK293 cells (mock) or stably expressing human MT_1_ or MT_2_ was performed using the p-ERK1/2 AlphaLISA Sure-fire ultra-Assay kit (PerkinElmer, Waltham, MA, USA) as described [32]. Briefly, cells were starved overnight before being stimulated with a saturating or increasing concentrations of melatonin or plant extracts for 5 min at 37 °C 5% CO_2_, collected in lysis buffer, and 4 µL of each sample was used for the assay according to the supplier’s instructions. The plate was read using the Infinite M1000 Tecan microplate reader. Data were analysed using GraphPad Prism software (La Jolla, CA, USA) and fitted via the non-linear regression to determine the EC_50_ values.

### 2.7. β-arrestin2 Recruitment Assay

The PathHunter assay from the DiscoverX company was used as described [33] to measure the recruitment of β-arrestin2 to melatonin receptors (DiscoverX/Eurofins). Briefly, HEK 293 parental cells stably expressing a fusion protein of β-arrestin2 and the larger N-terminal deletion mutant of β-gal were transiently transfected with human MT_1_ or MT_2_ receptors fused at their C-terminal part with the small enzyme fragment ProLink tag. Two days after transfection, the cells were stimulated with a saturating or increasing concentration of MLT or extracts for 2–3 h at 37 °C 5% CO_2_. β-arrestin2 recruitment resulted in the complementation of the two enzyme fragments and the formation of an active β-gal enzyme. Luminescence signals were determined after 60 min incubation at room temperature. Data were fitted via non-linear regression to determine EC_50_ values and normalized to basal levels using GraphPad Prism software.

### 2.8. BRET β-arrestin2 Recruitment Assay

For BRET assays, phenol red-free medium was removed from HEK293T cells and replaced by PBS that contained calcium and magnesium, as previously described [34]. The assay was carried out at room temperature and started by adding 10 µL coelenterazine h to the well to yield a final concentration of 5 µM. β-arrestin2 recruitment activity of **Ex19** (*Pistacia vera*) and melatonin was measured 5 min after addition of the Rluc substrate, and plate reading was performed 5 min after. Readings started 10 min after addition of the Rluc substrate. BRET signal was determined by calculating the ratio of the light emitted at 515–555 nm over the light emitted at 465–505 nm using a Mithras LB940 instrument (Berthold, Bad Wildbad, Germany) and filters with the appropriate band pass. Net BRET signals were determined by subtracting the BRET signal obtained with cells that only expressed Rluc-MT_1_ receptor from BRET signals obtained with cells that co-expressed each receptor and YFP-tagged β-arrestin-2 (this plasmid construct have never been described before). Curves were fitted using a nonlinear regression and log (agonist) vs. response fit equation using GraphPad Prism 9 (GraphPad Software, La Jolla, CA, USA). 

### 2.9. Determination of Melatonin Content by ELISA

The Melatonin ELISA SALIVA assay is based on the competition between a biotinylated and non-biotinylated antigen for fixed number of antibody binding sites that are already coated on the wells by the manufacturer [35]. After incubation 16–20 h at 2–8 °C, the wells are washed to stop the competition reaction. Once the substrate reaction has occurred, the intensity of the colour generated is inversely proportional to the amount of antigen in the sample. Six different concentrations of pure standards already prepared by the manufacturer (0, 0.5, 1.5, 5, 15, and 50 pg/mL) were used in duplicate to obtain a calibration curve. The melatonin ELISA kit was purchased from IBL TECAN company, and it also provided two different controls of known melatonin concentration as positive control, to check the efficiency and the validity of the assay.

### 2.10. MTT Assay

Cell viability was measured using the colorimetric 3-(4,5)dimethylthiazol-2-yl)-2,5 diphenyltetrazolium bromide (MTT) assay kit (Millipore Corporation, Billerica, MA, USA). Living cells reduce MTT to formazan which was quantified by measuring the absorbance at 570 nm. HEK293 cells expressing MT_1_ receptors were grown to confluence in 96-well plates and incubated with 100 µL of medium, DMSO or fixed concentration of extracts for 3 or 24 h at 37 °C 5% CO_2_. At the end of the incubation time, 20 µL MTT was added to each well and incubate for another 4 h at 37 °C 5% CO_2_. Formazan production was expressed as a percentage of the values obtained from control cells. The plate was read using an Infinite M1000 Tecan microplate reader. Data were analysed using GraphPad Prism software (La Jolla, CA, USA).

## 3. Results

### 3.1. Identification of Plant Extracts Interfering with 2─[^125^I]iodomelatonin Binding to Human MT_1_ and MT_2_ Receptors

We selected 25 plant extracts based on their reported beneficial properties in sleep disorders and anxiety (see Table 1 in Methods Section for more details). Pilot experiments determined that most extracts are soluble in the standard TEM binding buffer (see Materials and Methods) at a maximal concentration of 16 mg/mL. The remaining extracts (**Ex9**, **10**, **14**, **15**, **18**, **20**, and **25**) were dissolved at the same concentration in DMSO. Then, the capacity of different extract dilutions to interfere with 2─[^125^I]─MLT binding to human MT_1_ and MT_2_ in HEK293 cells expressing either receptor was determined. For those extracts diluted in DMSO, the maximal DMSO concentration was 1% which did not interfere with 2─[^125^I]─MLT binding. Most of the plant extracts contained between 20 and 50% maltodextrin as a major additive to increase the fluidity of the extract. Control experiments showed that maltodextrin alone in the relevant concentration range had no impact on 2─[^125^I]─MLT binding (Appendix A). As affinities cannot be determined in complex plant extracts, we determined the ‘apparent’ K_i_ of each extract by taking into account the IC_50_, the radioligand concentration and the respective K_d_ for MT_1_ and MT_2_ according to the Cheng–Prusoff equation (see Materials and Methods). Most extracts inhibited 2─[^125^I]─MLT binding with an apparent K_i_ ranging between 0.03 and 16 mg/mL (Table 2). Assuming an average molecular weight of 500 Da for small molecule constituents, this would translate into 60 µM to 32 mM concentrations (Figure 1A,B and Table 2). **Ex18** stood out with apparent K_i_ values of 3 and 10 ng/mL for MT_1_ and MT_2_ or an estimated molar concentration of 6 and 19 nM, respectively (Figure 1A,B and Table 2). Pure melatonin served as a standard with K_i_ values of 0.2 and 0.4 nM for MT_1_ and MT_2,_ respectively. Some extracts showed at least 10-fold selectivity for MT_1_ (**Ex23**, and **24**) others for MT_2_ (**Ex7**, **12**, and **15**) (Figure 1C,D and Table 2). For three of the extracts (**Ex13**, **17**, and **24**), a partial competition was observed with estimated I_max_ values between 32 and 42%, suggesting a more complex allosteric binding mode (Figure 1E,F and Table 2). All other extracts showed full displacement in the tested concentration range (5 × 10^−13^ to 32 mg/mL). Taken together, most of the 25 tested extracts inhibited 2-[^125^I]─MLT binding in a competitive manner with estimated apparent K_i_ values in the nM to mM range suggesting the existence of active principles in these extracts. Based on the apparent K_i_, **Ex6**, **Ex7**, **9**, **10**, **12**, **15**, **17**, **18**, **20**, and **25** were selected for further characterization in functional assays.

### 3.2. Characterization of Plant Extracts in Functional Melatonin Receptor Assays

Melatonin receptors are prototypical G_i/o_ protein-coupled receptors that inhibit adenylyl cyclases [11]. As the method to determine cAMP levels is based on an optical readout (fluorescence resonance energy transfer, FRET) and the plant extracts are colourful at high concentrations, the maximal extract concentration compatible with the assay was first defined. A maximal concentration of 0.16 mg/mL was selected (Appendix A). Then, the effect of the extracts on cell viability was evaluated at this concentration. A 3 h incubation was chosen as this corresponds to the longest incubation time used in our functional assays (Appendix A). Only **Ex7** showed a significant decrease in cell viability. Some extracts increased cell viability at 3 h but this effect was not seen after 24 h of incubation.

Afterwards, the effect of the selected extracts was examined on mock-transfected HEK293 cells. As expected, melatonin did not show any effect in these cells. Most of the extracts had a tendency in mock cells to reduce forskolin (FSK)-stimulated cAMP levels, with **Ex9**, **10** and **25** reaching statistical significance (Figure 2A). This effect was not due to the presence of maltodextrin as ruled out in control experiments (Appendix A). Then, the same experiment was repeated in HEK293 cells expressing either MT_1_ or MT_2_ receptors (Figure 2B,C). The data presented are normalized to mock cell responses to focus on melatonin receptor-dependent effects. Melatonin (1 µM) decreased FSK-stimulated cAMP production as expected. **Ex9**, **15**, **17**, **18** and **19** (0.16 mg/mL) showed a similar effect in cells expressing MT_1_. In cells expressing MT_2_, only **Ex15** and **Ex18** showed an effect with similar amplitude to melatonin, whereas **Ex7** and **Ex17** had a partial effect. Three extracts (**Ex9**, **15**, and **18**) were selected for further concentration–response experiments performed in parallel in mock cells and receptor expressing cells. **Ex9** showed a concentration-dependent decrease in FSK-stimulated cAMP production in MT_1_ expressing cells, which was similar to the effect observed in mock cells (Appendix A). A similar pattern was observed in cells expressing MT_2_ vs. mock cells (Appendix A). For **Ex15,** the concentration–response curve was left-shifted as compared to mock cells with an EC_50_ of 0.011 ± 0.011 mg/mL for MT_1_ expressing cells, confirming the presence of a MT_1_-dependent active principle in this extract (Figure 3A). No MT_2_-dependent effect could be revealed for **Ex15** (Figure 3B). The EC_50_ values obtained for **Ex18** were similar for both receptors (6.7 ± 5.70 × 10^−6^ mg/mL and 5.68 ± 5.23 × 10^−6^ mg/mL for MT_1_ and MT_2_, respectively) and clearly different from the mock cells (Figure 3C,D). The effect of **Ex18** on MT_2_ was reversed by the melatonin receptor antagonist luzindole (Figure 3E) and by pertussis toxin (PTX) pre-treatment (Figure 3F), confirming that the active principle of **Ex18** acts through the orthosteric binding site of melatonin receptors and activates G_i/o_ proteins as melatonin. Similar results were obtained when MT_1_ expressing cells were pre-treated with PTX (Appendix A). Taken together, several extracts displaying melatonin receptor-independent effects in the cAMP inhibition assay on mock cells reduced the observation window in cells expressing MT_1_ or MT_2_. This might be the reason why the effect of **Ex9** observed at a single concentration was not confirmed in concentration–response experiments. **Ex15** showed a modest MT_1_-dependent effect whereas **Ex18** showed a robust effect on both MT_1_ and MT_2_. 

The effect of the extracts was then evaluated on the ERK1/2 pathway, an integrative signalling pathway activated by most GPCRs, including melatonin receptors [32]. In mock cells, the reference compound melatonin was without effect, as expected, and several extracts (**Ex7**, **9**, **10**, and **15**) had a very strong effect (50–100-fold increase) (Figure 4A). For these extracts, it was impossible to observe a melatonin receptor-dependent effect in the cell lines expressing MT_1_ and MT_2_. Melatonin receptor-dependent increases (up to 12-fold) were seen for **Ex6**, **17**, **18**, **19**, and **25** (Figure 4B,C). This effect was replicated in concentration–response experiments performed in parallel in mock cells and receptor expressing cells (Figure 5A–F). EC_50_ values could only be determined for **Ex18** (5.60 ± 4.70 × 10^−7^ mg/mL and 7.00 ± 5.80 × 10^−5^ mg/mL for MT_1_ and MT_2_, respectively) (Figure 5C,D). For the other extracts, the effect was only measurable at the highest extract concentration (0.16 mg/mL). Interestingly, **Ex17** seems to be selective for MT_1_ at this concentration (Figure 5A,B). 

Recruitment of β-arrestins is another important feature of GPCRs that is involved in receptor endocytosis, desensitization and signalling [36]. β-arrestin2 recruitment was measured with a previously described β-galactosidase (β-gal) complementation assay [33]. Neither melatonin nor any extract induced β-arrestin2 recruitment in cells expressing the β-arrestin2-β-gal acceptor fusion protein and the β-gal donor alone (not fused to melatonin receptors) (Figure 6A). In cells expressing the MT_1_-β-gal donor fusion protein, melatonin and **Ex18** induced β-arrestin2 recruitment (Figure 6B), while in cells expressing the MT_2_-β-gal donor fusion protein, melatonin and **Ex9**, **Ex17** and **Ex18** induced β-arrestin2 recruitment (Figure 6C). Concentration–response experiments confirmed the effect of **Ex18** with EC_50_ values of 2.32 ± 2.56 × 10^−5^ and 4.49 ± 5.24 × 10^−6^ mg/mL for MT_1_ and MT_2_, respectively (Figure 6D,E). Results for MT_1_ were confirmed with a second β-arrestin2 recruitment assay based on the BRET technology. **Ex18** showed an EC_50_ of 3.40 ± 1.22 × 10^−5^ mg/mL and all other extracts were negative in this assay (Appendix A).

### 3.3. Ex18 Contains High Amounts of Melatonin 

Melatonin has been shown to be synthesized by plants [37]. To know whether the observed effect of **Ex18** could be, at least in part, explained by the presence of melatonin in the extracts, the melatonin content was assessed using ELISA and HPLC/MS. A content of 4.0 ± 0.5 mg and 5.29 ± 0.46 mg/g extract (*n* = 3–4) was determined via ELISA and HPLC-MS, respectively. Therefore, the apparent K_i_ and pEC_50_ values of **Ex18** were re-calculated based on the actual melatonin content and compared with the values measured for synthetic melatonin (Figure 7 and Table 3). The pK_i_ values of **Ex18** for 2─[^125^I]-MLT competition at MT_1_ and MT_2_ were significantly left-shifted (~1 log for MT_1_ and ~0.5 log for MT_2_) as compared to melatonin (Figure 7A,B and Table 3). A similar significant left-shift of pEC_50_ values of **Ex18** was observed in the cAMP and ERK1/2 assays in the range of 1 to 1.5 logs (Figure 7C–F and Table 3). Interestingly, the concentration–response curves of **Ex18** and melatonin for β-arrestin2 recruitment to MT_1_ and MT_2_ were superimposed (Figure 7G,H and Table 3). This observation was also confirmed in the BRET-based β-arrestin2 recruitment assay for MT_1_ (Appendix A). In sum, these results are compatible with the presence of biologically active melatonin in **Ex18** and in addition, suggest the presence of an additional active principle that either directly activates the receptor or potentiates the effect of endogenous melatonin. This effect seems to be limited to G_i_ protein-dependent effects and absent at the level of β-arrestin2 recruitment.

### 3.4. Ex18 Potentiates the Effect of Exogeneous Melatonin on the cAMP and ERK1/2 Pathways

To further elaborate on the possibility of a potentiation of the melatonin response, the question has been raised on whether the effect of exogenous melatonin can be also potentiated by **Ex18**. This issue was addressed using the ERK1/2 assay, which showed the most robust potentiation effect of **Ex18**, and a concentration of 10 nM of melatonin. For both receptors, the effect of exogenous melatonin was amplified by increasing concentrations of **Ex18** (Figure 8). For MT_1_, a left-shift of 1 log in pEC_50_ was observed, which, however, did not reach statistical significance (10.9 ± 0.76 M; *n* = 3) (*p*-value *p* = 0.45, paired *t*-test) (Figure 8A). In addition, for MT_2_, the pEC_50_ value was significantly left-shifted by three logs (12.6 ± 0.18 M; *n* = 3) (*p*-value ** *p* < 0.01, paired *t*-test) (Figure 8B) compared to the pEC_50_ value of **Ex18** alone. Collectively, these results support the hypothesis of the presence of a potentiator of melatonin in **Ex18**. 

Next, the possibility of a potentiation of the melatonin response at the cAMP pathway was investigated at a saturating (10 nM) and a threshold (0.1 nM) melatonin concentration for both melatonin receptors (Figure 9). At 0.1 nM melatonin, no potentiating effect of **Ex18** was observed for both melatonin receptors (Figure 9A,B). At 10 nM melatonin, **Ex18** was unable to further improve the melatonin response at MT_1_ receptors (Figure 9C) but amplified the effect of melatonin on MT_2_ receptors with a left-shifted (1.7 log) pEC_50_, which, however, did not reach statistical significance (11.1 ± 0.76; *n* = 3) (*p*-value *p* = 0.17, paired *t*-test) compared to **Ex18** alone (9.38 ± 0.78; *n* = 3) (Figure 9D). These results confirm the presence of a potentiator of melatonin in **Ex18** that is also measurable at the cAMP pathway.

### 3.5. Presence of Active Principle Potentiating the Effect of Exogeneous Melatonin in Several Plant Extracts

Based on the results of **Ex18**, the other extracts were also tested for the presence of a potentiator of exogenous melatonin on MT_1_ and MT_2_ receptors. We chose the ERK1/2 assay, with an EC_20_ concentration of melatonin (1 nM) and an extract concentration of 0.0016 mg/mL, which is l00 times lower than that used in Figure 4, to minimize the direct effect of the extracts alone on the ERK1/2 pathway (Figure 10). For MT_1_ receptors, only **Ex24** showed a statistically significant increase in the melatonin effect (Figure 10A,C,E). For the MT_2_ receptor, **Ex6**, **Ex8**, **Ex19**, and **Ex24** showed significant potentiating effects (Figure 10B,D,F). Then, the melatonin content for these four extracts was determined using HPLC/MS to estimate the melatonin amount that we might have added with the extracts in the ERK1/2 assay (Table 4). Despite the presence of various amounts of melatonin in the different extracts, for none of them the estimated concentration was high enough to activate melatonin receptors in our ERK1/2 assay. Collectively, several extracts seem to contain an active principle, different from melatonin, that potentiates the response of exogenous melatonin, in particular for the MT_2_ receptor at the ERK1/2 pathway.

## 4. Discussion

In the present study, we evaluated the capacity of 25 plant extracts to bind to melatonin MT_1_ and MT_2_ receptors, to induce signalling through the G_i_/cAMP and ERK1/2 pathways and to recruit β-arrestin2 in a melatonin receptor-dependent and -independent manner. A remarkable high number of extracts inhibited 2─[^125^I]─MLT binding to MT_1_ and MT_2_, although with a large range of apparent K_i_ values (typically in the high µM to mM range). Among the 10 plant extracts selected for signalling studies based on their apparent K_i_ values, **Ex18** from *Pistacia vera* dried fruits was active on both melatonin receptors in all functional assays with apparent EC_50_ values in the subnanomolar range. Five and four other extracts were active on the G_i_/cAMP and ERK1/2 pathways, respectively, in melatonin receptor expressing cells but either only at the highest concentration tested or with significant effects in mock cells. The high melatonin levels detected in **Ex18** are likely to contribute to the activity of **Ex18** on melatonin receptors. In addition, **Ex18** seems to have a potentiating property of exogenous melatonin and most likely of endogenous melatonin. This effect seems to be restricted to G_i/o_-dependent pathways and particularly visible in the ERK1/2 assay. Several other plant extracts (*Melissa officinalis*, *Eleutherococcus senticosus*, *Rhodiola rosea*, and *Valeriana officinallis*) reported to have sleep-promoting properties seem to contain active principles potentiating the response of exogenous melatonin on human melatonin receptors as well.

Plant extracts are complex mixtures containing active principles that are composed of one or several active components acting through the orthosteric or allosteric binding sites with potentiating or inhibiting properties or by modifying the receptor conformation more indirectly by modulating the membrane environment. When interpreting the results of binding and functional assays, this complexity has to be kept in mind. Derived K_i_ and EC_50_ values should be defined as apparent values. In the 2─[^125^I]─MLT binding assay, a high number of extracts inhibited the binding of this melatonin receptor-specific radioligand. The apparent K_i_ values varied widely from 3 × 10^−6^ to 15 mg/mL, with 7 extracts having K_i_ values lower or equal to 0.1 mg/mL for at least one receptor. Most of the extracts inhibited 2─[^125^I]─MLT binding fully, which is compatible with a competitive binding mode. For **Ex18**, a high melatonin content of 4–5 mg/g of extract was detected, which is likely to largely contribute to the observed apparent K_i_ of this extract in the low nanomolar range. When determining the EC_50_ of plant extracts on general intracellular signalling pathways (cAMP production, ERK1/2 phosphorylation, and β-arrestin recruitment), the inclusion of mock-transfected cells is mandatory to exclude the effects triggered by other receptors expressed by the cell type of choice. In our case, 3 and 4 out of the 10 extracts tested in functional assays were able to inhibit FSK-stimulated cAMP production and to activate ERK1/2, respectively, in mock-transfected HEK293 cells. As HEK293 cells do not express functional melatonin receptors, these effects are likely to be mediated by the activation of other receptors and proteins of these two signalling cascades [38]. Among the extracts showing activity on the G_i_/cAMP pathways in mock cells, **Ex9** showed a further inhibition in MT_1_-expressing cells at 0.16 mg/mL. However, this effect was not confirmed in concentration–response experiments, most likely because of the strong mock effect (50% inhibition), which reduces the experimental window for further inhibition through MT_1_ receptors. In the ERK1/2 assay, four extracts showed a strong (50–100-fold over basal) activation in mock cells, which is approximately 10 times higher than the amplitude of melatonin receptor activation (2–10 fold over basal). For these four extracts no significant further increase in ERK1/2 activation was measured in melatonin receptor expressing cells. Masking of a possible melatonin receptor-dependent ERK1/2 activation cannot be excluded under these conditions. **Ex17**, **18**, and **19** tested positive in the ERK1/2 assay for at least one receptor. Whereas **Ex18** showed a clear concentration-dependent curve, **Ex17** and **Ex19** were only active at the highest concentration (0.16 mg/mL) with **Ex17** being an additional MT_1_-selective. In the β-arrestin2 PathHunter assay, no signal was observed in the control cells while melatonin and **Ex18** activated melatonin receptors in a concentration-dependent manner. 

Apparent pK_i_ values and pEC_50_ values for the cAMP and the ERK1/2 pathways of **Ex18** were unexpectedly left-shifted by 1–1.5 logs compared to pK_i_ and pEC_50_ values for melatonin, when considering the melatonin content of **Ex18** determined using ELISA and HPLC/MS. Three arguments indicate that this left-shift is not due to an erroneous determination of the melatonin content of **Ex18**: (i) two independent methods (ELISA, and HPLC/MS) were used; (ii) the shift was not observed for pEC_50_ values of β-arrestin2 recruitment in two independent assay formats; and (iii) the shift was also observed for exogenously added melatonin in **Ex6**, **8**, **19**, and **24** devoid of significant endogenous melatonin amounts. These results not only indicate the presence of biologically active plant-derived melatonin in **Ex18** but also suggest the presence of an active principle potentiating the effect of melatonin on its receptors in several plant extracts. Whereas we can exclude that **Ex6**, **8**, **19**, and **24** contain a melatonin receptor agonist (no effect on its own), we cannot formally exclude this possibility for **Ex18** which would exist in addition to melatonin in this extract. However, the hypothesis of a second agonist is rather unlikely as it should have an even higher affinity than melatonin itself to explain the left-shift in the context of a high endogenous melatonin content. In addition, structurally similar molecules, such as melatonin precursors or metabolites, are unlikely candidates as they show typically lower affinities for melatonin receptors. Further possible explanations, like a synergistic effect between melatonin through melatonin receptors, and other molecules present in the extract, acting on receptors or components of the signalling pathway monitored, are also among other action mechanisms. The absence of the effect of **Ex6**, **18**, and **19** in mock cells in all functional assays argues against the presence of a second melatonin receptor-independent active component acting in synergy with melatonin receptors. Therefore, the hypothesis of the presence of a PAM that potentiates the effect of melatonin (endogenous and exogenous) was favoured. Interestingly, the potentiating effect seems to be restricted to G protein-dependent signalling events as absent at the level of β-arrestin2 recruitment but visible on the G_i_/cAMP pathway and the ERK1/2 pathway which is G_i/o_-dependent and β-arrestin-independent in HEK293 cells [32]. The fact that the potentiating effect is particularly visible at the level of ERK1/2 activation can be most likely explained by the highly integrative character of this pathway that leads to a substantial signal amplification. However, at the current state, we cannot formally exclude the possibility that our β-arrestin assays are less sensitive in detecting PAMs.

As there are no allosteric modulators currently known for melatonin receptors, it is impossible to make any predictions concerning the structure of these putative PAMs. Comparison of the identified active ingredients reported by the extract suppliers in the extracts containing a potentiating activity (**Ex6**, **Ex8**, **Ex19**, and **Ex24**) with the literature allows to establish only loose connections. Rosmarinic acid present in **Ex6** (*Melissa officinalis)* has been shown to have antidepressant properties [39], eleutheroside B and E present in **Ex8** (*Eleutherococcus senticosus*) show antidepressant activity in animal models [40,41] and rosavins and salidroside present in **Ex19** (*Rhodiola rosea*) show antidepressant and anxiolytic properties [42]. The real chemical nature of these melatonin receptor PAMs would have to be determined after isolation from plant extracts. Two alternative approaches can be envisioned: (i) sequential fractionation of plant extracts followed by functional testing in melatonin receptor expressing cells can be performed to isolate the PAM activity of extracts; (ii) an affinity mass spectrometry-based screening recently employed to identify a new agonist of serotonin 2C receptor from plant extracts [5]. The latter is based on the capacity of the receptor to maintain its binding property upon receptor solubilization and immobilization on an affinity column and the confirmation of binders in functional assays. 

Potentiation has been described before in plant extracts. It can exist between two compounds of a single plant extract, two compounds from two different plant extracts, or between a compound and a synthetic drug [43]. In extreme cases, the potentiating or combinatorial effect is only observed by the mixture of compounds present in a plant extract, whereas each ingredient in isolation lacks therapeutic activity [44]. 

*Pistacia vera* (**Ex18**) is a member of the *Anacardiaceae* family and is an important medicinal plant with various reported pharmacological functions [45]. Several types of phytochemical constituents like terpenoids, phenolic compounds, fatty acids, and sterols have been isolated and identified from different parts of *Pistacia* species, mostly of Iranian origin. Accidentally, *Pistacia* species are among the plants containing the highest melatonin levels reported [37,46]. Whereas the melatonin content of plants is typically in the range of 1–100 ng per gram of plant material [47], 200 µg (and higher) of melatonin per gram have been reported in *Pistacia* [48]. However, there seems to be a considerable variability in the melatonin content of *Pistacia* samples as mentioned in the literature [49]. This might be due to differences in the season or location of harvest, in the extraction procedure or the detection method, as observed for many natural extracts [50]. Our ELISA and HPLC/MS data consistently indicate that **Ex18** contains high melatonin levels of around 5 mg/g extract. As the **Ex18** used in our study is an extract of dried *Pistacia* seeds (syn. Kernel) without additives, such as maltodextrin, the amount per extract matches the amount per dried kernel. Importantly, the melatonin present in **Ex18** is of natural origin excluding presence of any fossil fuel/synthetic melatonin based on a radiocarbon analysis performed by an independent analytic testing laboratory on the request of the supplier (personal communication). Collectively, these data confirm the high levels and the natural origin of melatonin in **Ex18**. Given the high variability of melatonin levels in *Pistacia* species reported in the literature, our data cannot be necessarily extrapolated to other *Pistacia* species and extracts. A larger follow-up study with a number of different *Pistacia* extracts will be necessary to determine the range of melatonin content in *Pistacia* extracts with standardized up-to-date analytical methods.

The typical amount of melatonin available in prescribed or over-the-counter medications for human use is 1–3 mg/day. The consumption of plants containing typical melatonin concentrations (nanograms per gram of plant material) is unlikely to have any physiological effect in humans. In contrast, the consumption of 0.5–1 g of the *Pistacia vera* extract tested in our study would be sufficient to ingest 2–5 mg of melatonin and to trigger a physiological effect in humans. 

Melatonin is extensively used to promote sleep initiation and for better adjustment of circadian misalignment as experienced when traveling over several time zones. Melatonin is also used as a prophylactic anti-ageing treatment and as a preventive treatment for neurodegenerative diseases and cancer [51,52]. As the amplitude of the natural melatonin rhythm tends to decline with age and in many disease states, this can be seen as a compensatory strategy. Our findings open new conceptual perspectives to potentiate the effect of melatonin, either by potentiating low endogenous levels in humans or by potentiating the effect of exogenous melatonin. Apart from lowering the dose of exogenous melatonin in combination with extracts containing a PAM, an additional advantage of the melatonin/plant extract combination might reside in the presence of additional beneficial effects of the plant extract.

## 5. Conclusions

In the current study, we identified high melatonin levels in a *Pistacia* extract among 24 other extracts. This melatonin is biologically active on MT_1_ and MT_2_ receptors in terms of binding and signal transduction. The *Pistacia* extract characterized in this study contains a potentiating activity of the endogenous melatonin but also of exogeneous melatonin. A similar potentiating activity was identified in four other extracts in the presence of exogenous melatonin. This study supports previous studies suggesting high melatonin levels in dry fruits of *Pistacia* and opens new perspectives for the presence of plant-derived allosteric modulators for GPCRs that potentiates the response triggered by an orthosteric ligand.

## Figures and Tables

**Figure 1 pharmaceutics-15-01845-f001:**
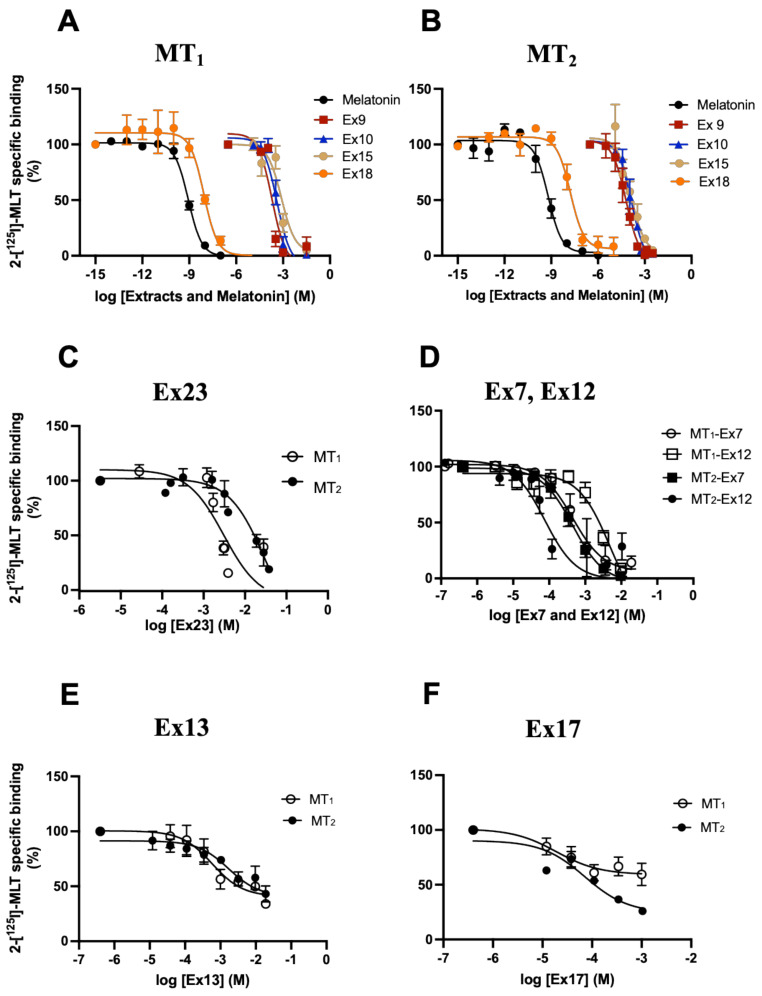
Competition of 2─[^125^I]─MLT binding by melatonin and plant extracts. Crude membrane preparations of HEK293 cells stably expressing MT_1_ (**A**) or MT_2_ (**B**) receptors were incubated with 100 pM of 2─[^125^I]-MLT and increasing concentrations of the indicated extracts. Competition curves of extracts with MT_1_ (**C**) and MT_2_ (**D**) selectivity (white symbols (MT_1_), black symbols (MT_2_)). Competition curves of extracts reaching a plateau at partial competition levels (**E**,**F**) (white symbols (MT_1_), black symbols (MT_2_)). The molarity of the active molecules present in extracts was estimated assuming an average molecular weight of 500 Da for small molecule constituents. Data are expressed as mean ± SEM of 3 independent experiments performed in duplicates. Curves were analysed using non-linear regression. Log EC_50_ values were compared for receptor selectivity for **Ex23** (**C**) and **Ex7** and **Ex12** (**D**) via the extra sum-of-squares F test. *p*-values for differences in Log EC_50_ were *p* < 0.0001 for **Ex23** (**C**); 0.0022 and 0.0001 for **Ex7** and **Ex12**, respectively. The bottom of the curves was compared for the partial competition for **Ex13** (**E**) and **Ex17** (**F**). *p*-values for differences in bottom were 0.35 and 0.0012 for **Ex13** and **Ex17**, respectively.

**Figure 2 pharmaceutics-15-01845-f002:**
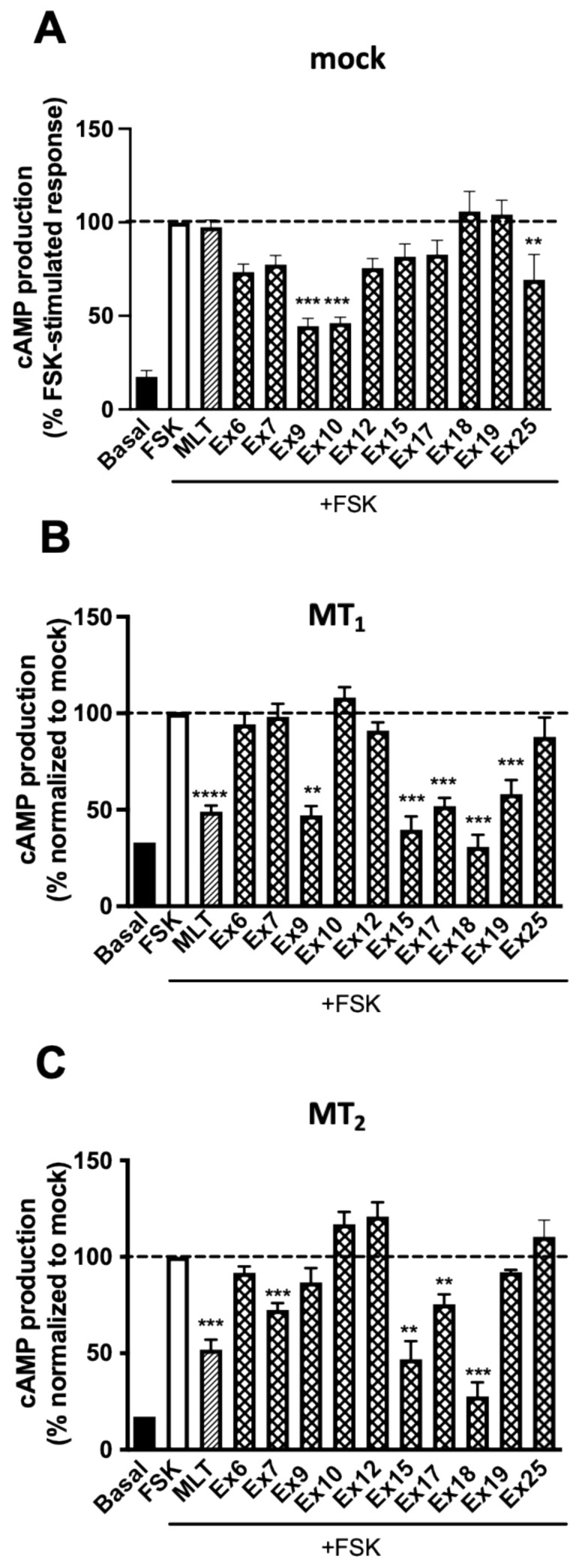
Effect of selected plant extracts on forskolin-stimulated cAMP production. Inhibitory effect of selected plant extracts on forskolin-stimulated cAMP production in HEK293 empty vector transfected cells (mock) (**A**) or cells stably expressing MT_1_ (**B**) or MT_2_ (**C**) receptors. Extracts were tested at a saturation concentration (3.2 × 10^−4^ M i.e., 0.16 mg/mL, except **Ex18**, which was tested at 8.69 × 10^−8^ M, equivalent to 0.005 mg/mL) and melatonin (1 µM). Data are normalized to maximal effect of forskolin-stimulated response (** *p* < 0.01; *** *p* < 0.001. (**A**) One-way ANOVA, *n* = 10; (**B**,**C**) two-tailed Student’s *t*-test, (** *p* = 0.014; *** *p* < 0.001; **** < 0.0001. *n* = 3. All experiments were performed in triplicates.

**Figure 3 pharmaceutics-15-01845-f003:**
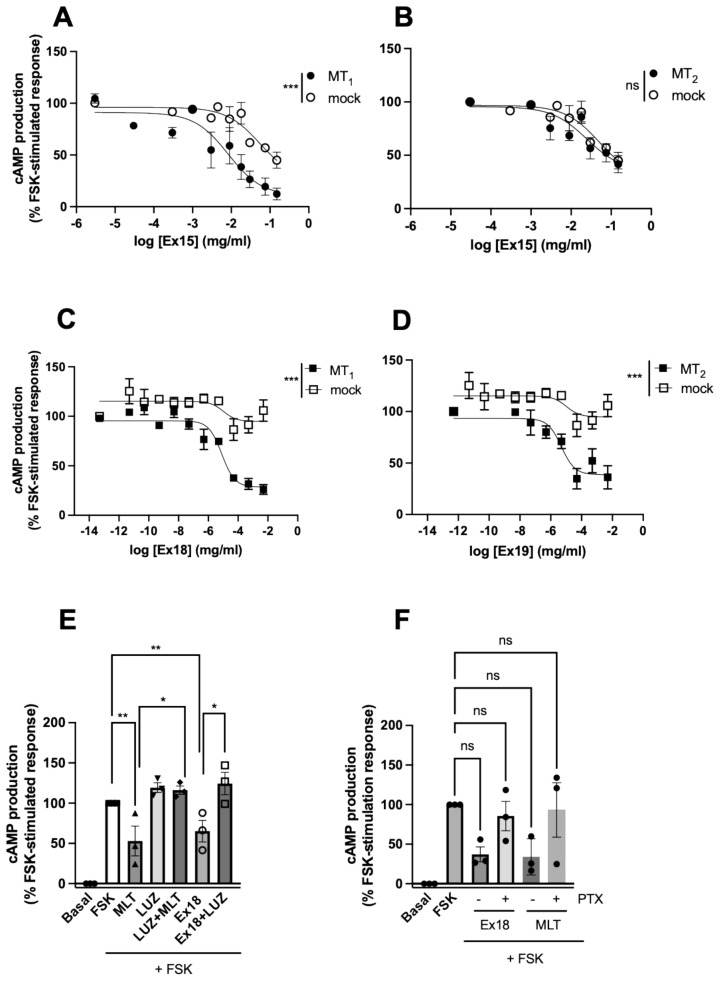
Effect of selected plant extracts on forskolin-stimulated cAMP production. Effect of increasing concentrations of **Ex15** and **18** on HEK293 cells transfected with empty vector (mock), MT_1_ (**A**,**C**) or MT_2_ (**B**,**D**) receptors. Data are expressed as mean ± SEM of 3 experiments performed in triplicates. Curves were analysed using non-linear regression. The log EC_50_ and bottom values were compared for receptor-dependent effect using extra sum-of-squares F test. ** *p* < 0.001 and *** *p* < 0.0001. (**E**) Reversion of the effect of *Pistacia vera* in cAMP production by luzindole in HEK293 cells expressing MT_2_ receptor. Cells were stimulated with exogenous MLT (10 nM) and **Ex18** (equivalent to 1 nM endogenous MLT in **Ex18**) in the presence or absence of luzindole (10 µM). Data are normalized to maximal effect of forskolin-stimulated response and represent the mean ± S.E.M. of 3 independent experiments performed in triplicates. One-way ANOVA * *p* < 0.05; ** *p* < 0.01. (**F**) G_i_ protein-dependent effect of *Pistacia vera* in HEK cells expressing MT_2_ receptors. Cells were pre-treated or not overnight with *Pertussis toxin* (PTX) (10 ng/mL) and stimulated with MLT or **Ex18**. Data are normalized to maximal effect of forskolin-stimulated response and are presented as mean ± S.E.M. of 3 experiments performed in triplicates. One-way ANOVA (* *p* < 0.05, ns, not significant).

**Figure 4 pharmaceutics-15-01845-f004:**
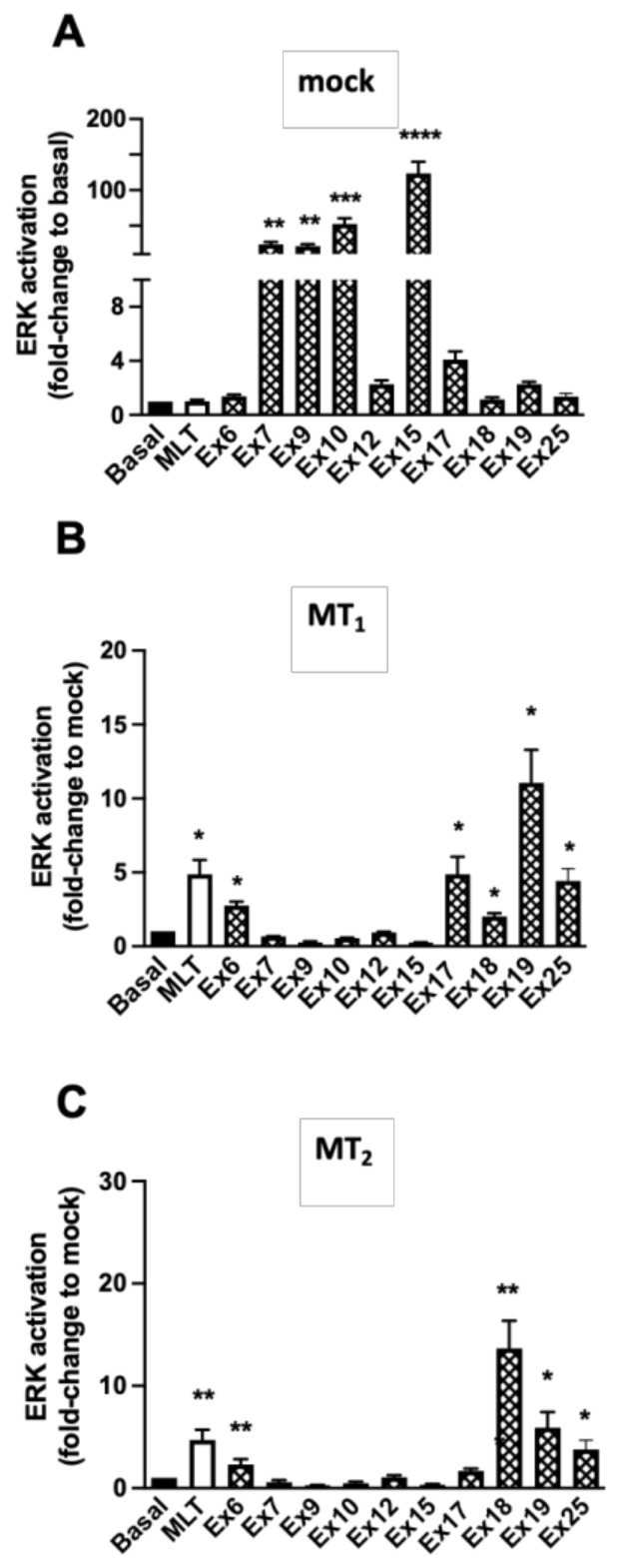
Effect of selected plant extracts on phospho-ERK levels. Detection of ERK1/2 phosphorylation in HEK293 empty vector transfected cells (mock) (**A**) or stably expressing MT_1_ (**B**) or MT_2_ (**C**) receptors. Extracts were tested at a saturation concentration (3.2 × 10^−4^ M i.e., 0.16 mg/mL, except **Ex18**, which was tested at 1.07 × 10^−7^ M, equivalent to 0.005 mg/mL) and MLT (1 µM). Data are expressed as mean ± SEM, *n* = 3–5). (**A**) (** *p* < 0.01; *** *p* < 0.0001; **** *p* < 0.0001; One-way ANOVA). (**B**) (* *p* = 0.02; two-tailed Student’s *t*-test); (**C**) (* *p* = 0.01; ** *p* = 0.004; two-tailed Student’s *t*-test).

**Figure 5 pharmaceutics-15-01845-f005:**
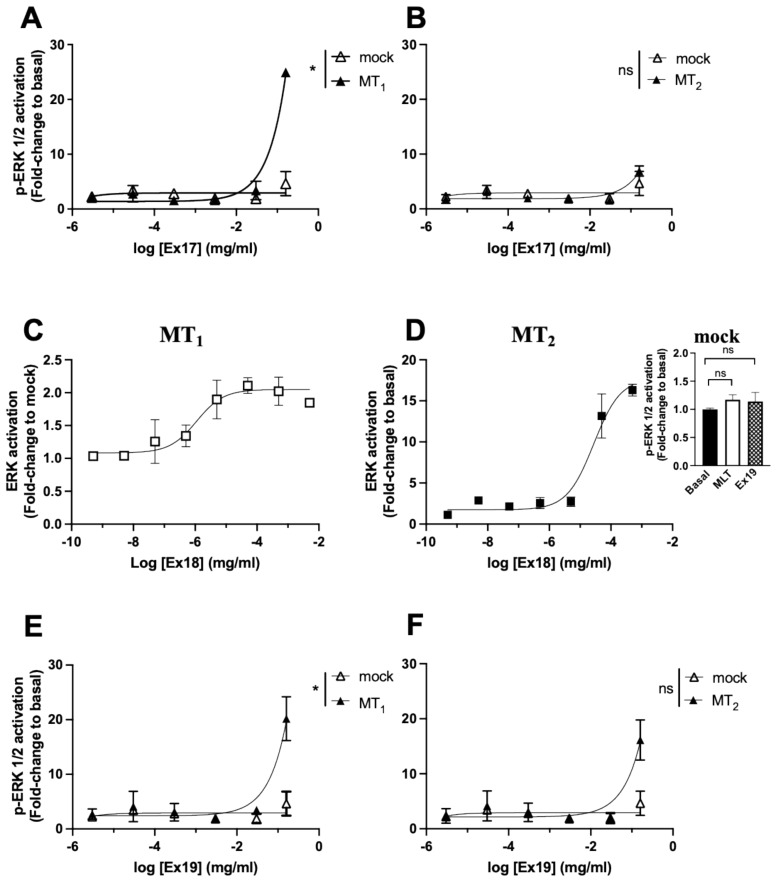
Concentration-dependence of the effect of selected plant extracts on phospho-ERK levels. Effect of increasing concentrations of **Ex17**, **18** and **19** on phospho-ERK1/2 levels in HEK293 empty vector transfected cells (mock) vs. HEK293 expressing MT_1_ (**A**,**C**,**E**) or MT_2_ receptors (**B**,**D**,**F**). Inset represents the effect of MLT and **Ex18** at the maximum concentration on mock cells. Data represent the mean of three independent experiments performed in duplicates. (**A**,**E**) (* *p* = 0.02; ns: not significant; two-tailed Student’s *t*-test).

**Figure 6 pharmaceutics-15-01845-f006:**
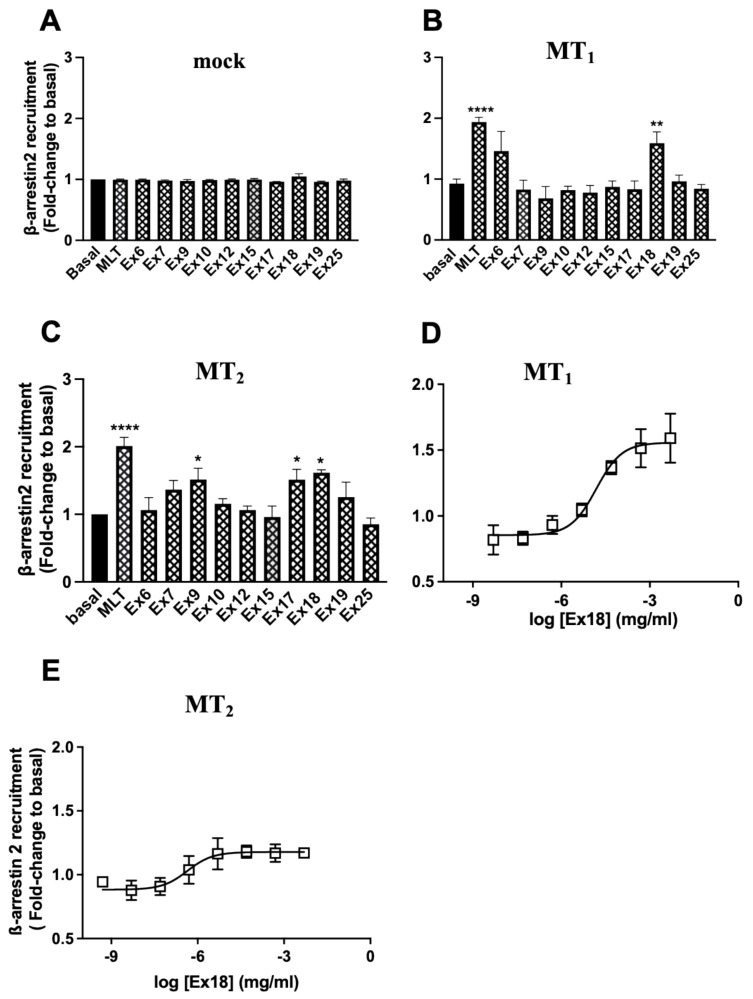
Effect of a saturating concentration of **Ex9**, **17** and **18** on β-arrestin2 recruitment assay. A single concentration of 3.2 × 10^−4^ M/0.16 mg/mL for extracts **9, 17** and **18**; 1.07 × 10^−7^ M/0.005 mg/mL for **Ex18** and MLT 1 µM was tested in HEK293 empty vector transfected cells (mock) (**A**) and HEK293 expressing MT_1_ (**B**) and MT_2_ (**C**) receptors. (**D**,**E**) show concentration–response curves of **Ex18** on HEK293 cells expressing MT_1_ and MT_2_ receptors, respectively. Data are expressed as mean ± SEM of 3 and 5 experiments respectively performed in triplicates. (* *p* < 0.05; ** *p* < 0.01; **** *p* < 0.0001; One-way ANOVA).

**Figure 7 pharmaceutics-15-01845-f007:**
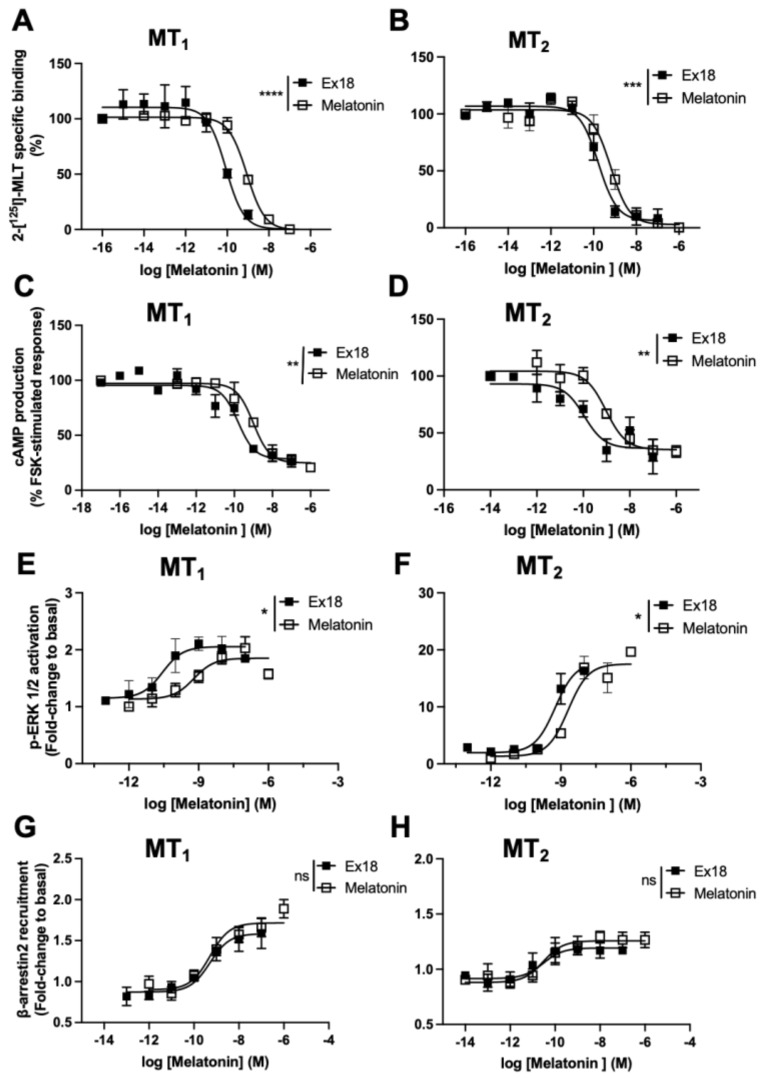
Replotting of experimental data obtained with **Ex18** (*Pistacia vera)* based on the actual melatonin amount of **Ex18** determined using HPLC-MS. (**A**,**B**) Competitive binding curves, concentration–response curve of the inhibition of cAMP production (**C**,**D**), ERK1/2 activation (**E**,**F**) and β-arrestin2 recruitment (**G**,**H**). Experiments with synthetic melatonin were run in parallel. Data are expressed as mean ± SEM, *n* = 3. Curves were analysed using non-linear regression. The log IC_50_ and EC_50_ values were compared between **Ex18** and MLT via the extra sum-of-squares F test. *p*-values for log IC_50_ were **** *p* < 0.0001 (**A**), *** *p* = 0.0008 (**B**) and for log EC_50_ ** *p* = 0.002 (**C**), ** *p* = 0.009 (**D**), * *p* = 0.07 (**E**), * *p* = 0.02 (**F**). ns, not significant (**G**,**H**).

**Figure 8 pharmaceutics-15-01845-f008:**
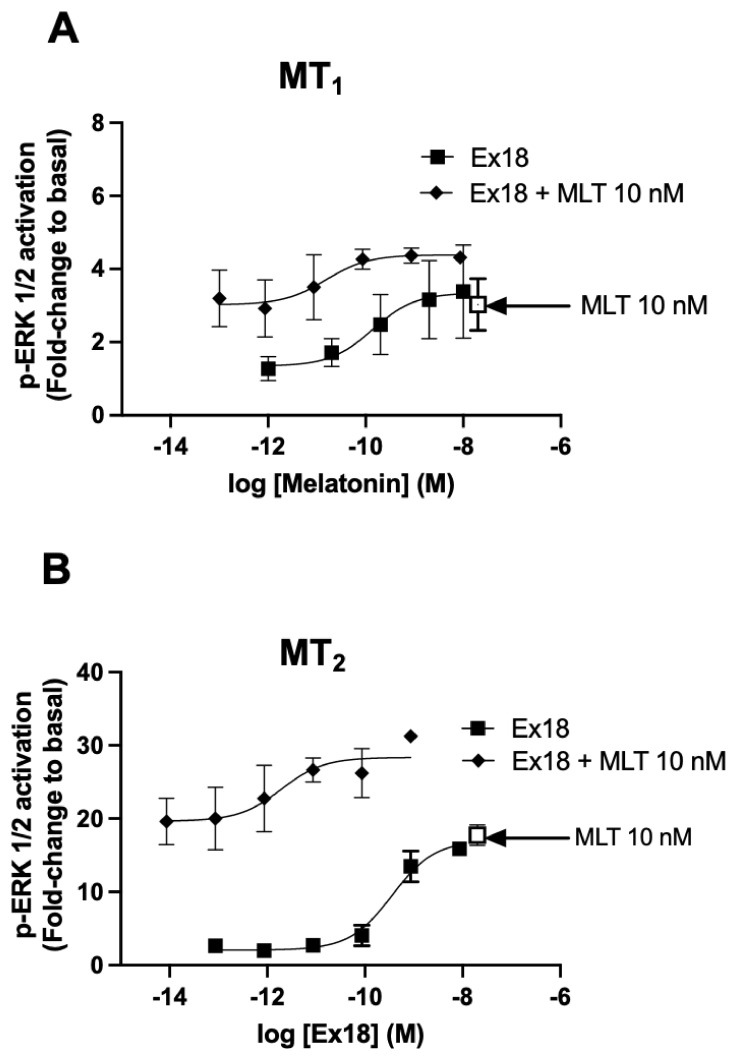
Potentiation of melatonin effect by *Pistacia vera* extract on ERK1/2 activation. Effects of increasing concentrations of MLT, **Ex18** alone or in presence of 10 nM MLT are shown on p-ERK levels in HEK293 cells expressing MT_1_ (**A**) or MT_2_ (**B**) receptors. The concentration–response curves of **Ex18** were calculated in molarity after having determined its content in MLT using HPLC-MS. Data are presented as mean ± SEM, *n* =3–4. Curves were analysed using non-linear regression. The E_max_ and Log EC_50_ values were compared between **Ex18** vs. **Ex18** + MLT (10 nM) using the extra sum-of-squares F test. *p*-values for E_max_ and log EC_50_ were *p* = 0.17, 0.47 for MT_1_ (**A**), respectively.

**Figure 9 pharmaceutics-15-01845-f009:**
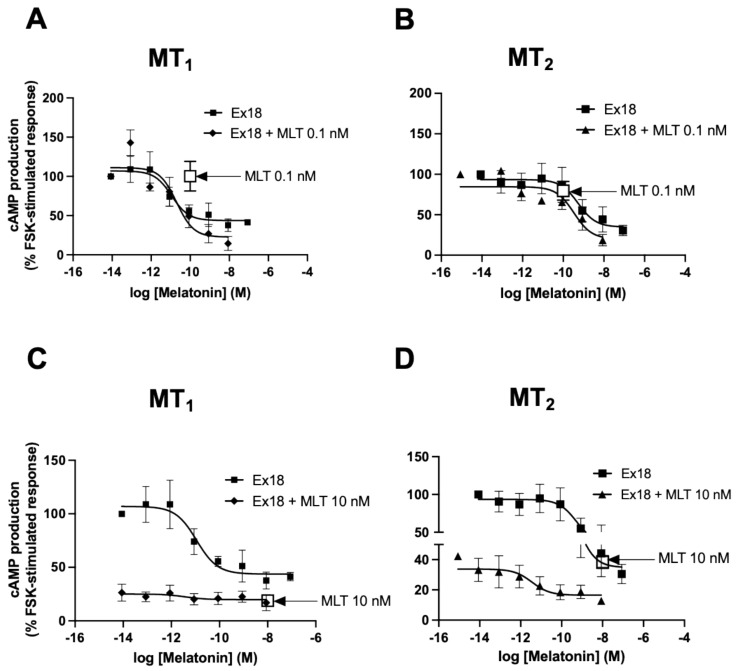
Potentiation of melatonin effect by *Pistacia vera* extract on cAMP production. Effects of increasing concentrations of MLT, **Ex18** alone or in presence of 10 nM (**A**) or 0.1 nM MLT are shown on cAMP production levels in HEK293 cells expressing MT_1_ and MT_2_ receptors. The concentration–response curves of **Ex18** were calculated in molarity after having determined its melatonin content by HPLC-MS. Data are presented as mean ± SEM, *n* = 3. Curves were analysed using non-linear regression. The log EC_50_ and E_max_ values were compared between **Ex18** vs. **Ex18** + MLT at 10 nM (**C**,**D**) and 0.1 nM (**A**,**B**) using the extra sum-of-squares F test. *p*-value for log EC_50_ were 0.69 (**B**) 0.63 (**D**); the *p*-value for E_max_ was 0.08 (**C**).

**Figure 10 pharmaceutics-15-01845-f010:**
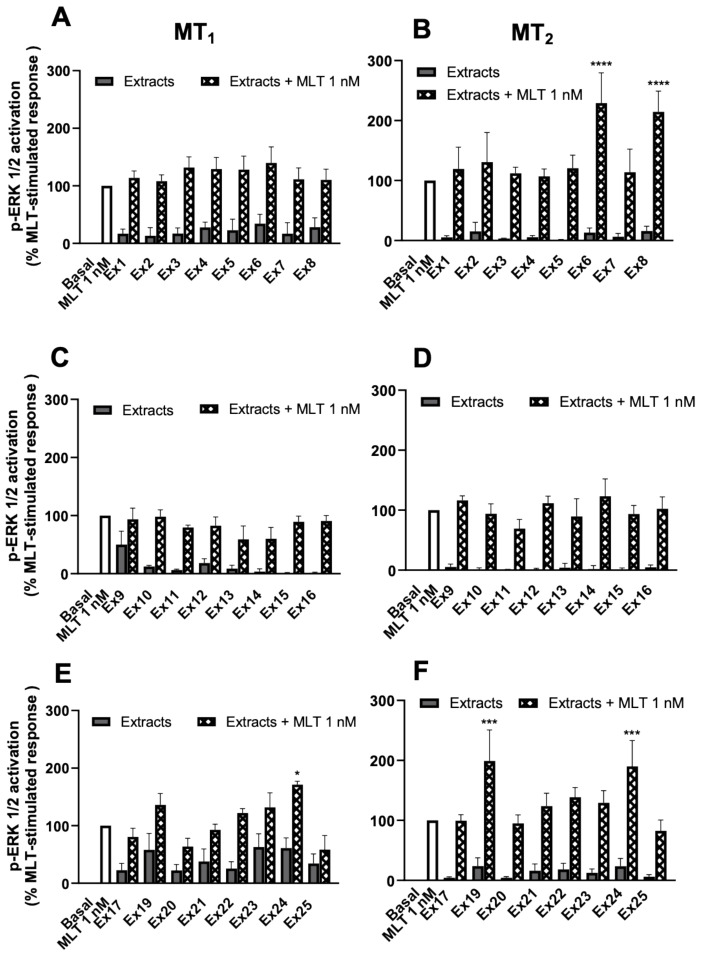
Potentiating effect of plant extracts on exogeneous melatonin on the ERK 1/2 pathway. Effect of 24 plant extracts tested at 0.0016 mg/mL alone or in the presence of melatonin (1 nM) is shown on p-ERK levels in HEK293 cells expressing MT_1_ (**A**,**C**,**E**) or MT_2_ (**B**,**D**,**F**) receptors. Data are presented as mean ± SEM, *n* = 3–5. * *p* < 0.05; *** *p* < 0.001; **** *p* < 0.0001. One-way ANOVA. All experiments were performed in duplicates.

**Table 1 pharmaceutics-15-01845-t001:** List of the plant extracts used in this study with their provider data and attribution code.

Code	Scientific Name	Plant Part	Extraction Method ^a^	Identified Active Ingredients/Extraction Techniques Used ^a^	Provider
**Ex1**	*Melissa officinalis*	Leaves	EtOH/H_2_O (30%/70%) *v/v*	Rosmarinic acid (7.02%)/Ultraviolet-visible-VIS	PLANTEX
**Ex2**	*Passiflora incarnata* L.	Aeral part	EtOH/H_2_O (30%/70%) *v/v*	Flavonoids; vitexin (4.18%)/Ultraviolet- visible-VIS	PLANTEX
**Ex3**	*Eschscholizia californica cham*	Flowering grass	EtOH/H_2_O	Californidine (0.5%)/-	BERNETT
**Ex4**	*Crocus sativus*	Red stigmas	H_2_O/EtOH (20%/80%)	Flavonoids: vitexin (2.99%)/HPLC	GREEN PLANT EXTRACT
**Ex5**	*Passiflora incarnata* L.	Aerial part	H_2_O (100%)	Flavonoids: vitexin (2.99%)/HPLC	NATUREX
**Ex6**	*Melissa officinalis* L.	Leaves	EtOH/H_2_O (30%/70%) *v/v*	Rosmarinic acid (7.08%)/Ultraviolet-visible-VIS	PLANTEX
**Ex7**	*Eschscholzia californica cham*	Flowering grass	EtOH/H_2_O	Californidine (0.45%)/-	BERNETT
**Ex8**	*Eleutherococcus senticosus*	Roots	EtOH/H_2_O (30%)	Eleutheroside B and E (0.36%)/-	HEALTH FOOD
**Ex9**	*Hypericum perforatum*	Unknown	EtOH	Flavonoids (10.5% rutin); hyperforin (3%)/HPLC	EUROMED
**Ex10**	*Matricaria chamomilla*	Flowers	EtOH/H_2_O (74%/26%)	Apigenin (1.61%)/HPLC	NATUREX
**Ex11**	*Zizyphus jujuba*	Seeds	EtOH/H_2_O (60–70%)	-	HUISONG
**Ex12**	*Humulus lupulus*	Flowers	EtOH/H_2_O (96%)	-	HUISONG
**Ex13**	*Withania somnifera*	-	-	Withaferin A (0.29%)/HPLC	NATREON
**Ex14**	*Griffonia simplicifolia*	Seeds	EtOH/H_2_O (40–60%)	5-HTP or L-5 Hydroxytryptophan (>30%)/HPLC	NATUREX
**Ex15**	*Crataegus monogyna*	Leaves and Flower	EtOH/H_2_O (80%/20%)	Flavonoids: vitexin-2-orhamnoside (1.80–3%)/HPLC	NATUREX
**Ex16**	*Asparagus officinalis*	Stems	-	-	AMINOUP
**Ex17**	*Camellia sinensis*	-	-	-	-
**Ex18**	*Pistacia vera*	Fruits, dry	EtOH/H_2_O (30%)	-	NETWORK NUTRITION
**Ex19**	*Rhodiola rosea* L.	Roots	EtOH/H_2_O (45–65%)/(35–55%)	Rosavins and salidroside (>1; >3%)/HPLC	NATUREX
**Ex20**	*Mentha x piperita* L.	Aerial part	EtOH/H_2_O (30%/70%)	-	NATUREX
**Ex21**	*Camellia sinensis* L.	Leaves	H_2_O (100%)	-	NATUREX
**Ex22**	*Malvaceae tilia cordata*	Flowers	H_2_O (100%)	-	PLANTEX
**Ex23**	*Verbena officinalis* L.	Leaves	H_2_O (100%)	-	PLANTEX
**Ex24**	*Valeriana officinallis* L.	Roots (rhizomes, and stallions)	H_2_O (100%)	Sesquiterpenic acids (Hydroxyvalerenic and acetoxyvalericic acid) (0.062%)/HPLC	EUROMED
**Ex25**	*Valeriana officinallis* L.	Roots (rhizomes, and stallions)	EtOH (70%)	Valerenic acid (0.3%)/HPLC	EUROMED

^a^, information from technical data sheet of provider; -, no data available.

**Table 2 pharmaceutics-15-01845-t002:** Inhibitory effect of extracts in the 2─[^125^I]iodomelatonin competitive binding assay on human MT_1_ and MT_2_ receptors.

Extract	MT_1_	MT_2_	MT_1_/MT_2_ Ratio
	App. K_i_ ± S.E.M (mg/mL) (µM)		
**Ex1**	0.63 ± 0.39 (*1260*)	0.69 ± 0.45 (*1380*)	0.91
**Ex2**	0.14 ± 0.07 (*280*)	0.09 ± 0.08 (*180*)	1.55
**Ex3**	0.20 ± 0.13 (*400*)	0.07 ± 0.04 (*140*)	2.85
**Ex4**	1.73 ± 0.42 (*3460*)	1.99 ± 1.20 (*3980*)	0.86
**Ex5**	0.13 ± 0.05 (*260*)	0.14 ± 0.41 (*280*)	0.92
**Ex6**	0.31 ± 0.13 (*620*)	0.16 ± 0.10 (*320*)	1.94
**Ex7**	0.49 ± 0.57 (*980*)	0.04 ± 0.04 (*80*)	12.25
**Ex8**	2.52 ± 2.33 (*5040*)	0.42 ± 0.28 (*840*)	6.00
**Ex9**	0.05 ± 0.01 (*100*)	0.03 ± 0.03 (*60*)	1.66
**Ex10**	0.21 ± 0.08 (*420*)	0.07 ± 0.04 (*140*)	3.00
**Ex11**	0.29 ± 0.08 (*580*)	0.36 ± 0.01 (*720*)	0.80
**Ex12**	2.04 ± 1.00 (*4080*)	0.20 ± 0.13 (*400*)	10
**Ex13** **Ex14**	~1.47 ± 1.54 (*2940*) ^b^ ~12.54 ± 13.84(*25,080)* ^a^	~0.59 ± 0.71 (*1180*) ^b^ 2.90 ± 1.64 (*5800*)	~2.49 ~4.32
**Ex15**	0.81 ± 0.51 (*1620*)	0.07 ± 0.04 (*140*)	11.57
**Ex16**	7.80 ± 1.09 (*15,600*)	5.68 ± 2.92 (*11,360*)	1.37
**Ex17**	~0.009 ± 0.004 (*18*) ^c^	~0.03 ± 0.02 (*60*) ^c^	~0.3
**Ex18**	3.31 ± 1.57 10^−6^ (*0.00662*)	9.88 ± 4.76 10^−6^ (*0.0196*)	0.33
**Ex19**	0.08 ± 0.07 (*160*)	0.19 ± 0.19 (*380*)	0.42
**Ex20**	0.44 ± 0.16 (*880*)	0.20 ± 0.11 (*400*)	2.2
**Ex21**	0.70 ± 0.44 (*1400*)	1.94 ± 2.54(*3880*)	0.36
**Ex22**	~9.38 ± 2.84 (*18,760*) ^a^	~3.44 ± 4.82 (*6880*) ^a^	~2.73
**Ex23**	~0.76 ± 0.25 (1520) ^a^	~5.07 ± 2.60 (*10,140*) ^a^	~0.15
**Ex24**	~1.22 ± 0.70 (*2440*) ^d^	~14.94 ± 14.67 (*29,880*) ^d^	~0.08
**Ex25**	0.18 ± 0.11 (*360*)	0.24 ± 0.06 (*480*)	0.75
**MLT**	0.11 ± 0.03 10^−6^ (*0.000224*)	0.20 ± 0.19 10^−6^ (*0.00040*)	0.56

The results are expressed as apparent K_i_ ± S.E.M. (*n* = 3) (K_d_ = 152 and 367 pM for MT_1_ and MT_2_, respectively). NC, no competition observed up to 16 mg/mL; n.d., not determined; ^a^, IC_50_ only estimated because no full inhibition observed at the highest extract concentration used (16 mg/mL 32,000 µM); ^b^, Imax = 34 ± 3% and 43 ± 10% for MT_1_ and MT_2_, respectively; ^c^, Imax = 59 ± 14%; 37 ± 16% for MT_1_ and MT_2_, respectively; and ^d^, Imax = 42% and 32% for MT_1_ and MT_2_, respectively.

**Table 3 pharmaceutics-15-01845-t003:** Functional characteristics of **Ex18** in HEK293 cells expressing melatonin receptors.

Receptor	2─[^125^I]─MLT Compet.		cAMP Inh.	ERK Act.	β-ARR
		pK_i_ ± S.E.M.	pEC_50_ ± S.E.M.		
**MT_1_**	Melatonin	9.27 ± 0.12	9.02 ± 0.05	8.73 ± 0.65	9.12 ± 0.16
	Ex18	* 10.2 ± 0.19	10.3 ± 0.96	** 10.4 ± 0.64	9.60 ± 0.54
**MT_2_**	Melatonin	9.27 ± 0.48	8.95 ± 0.39	8.71 ± 0.17	10.0 ± 0.87
	Ex18	9.79 ± 0.28	9.99 ± 0.34	* 9.38 ± 0.07	10.8 ± 0.99

Concentration–response curves were analysed using non-linear regression. Binding affinity was measured with 2─[^125^I]iodomelatonin and is expressed as mean apparent pK_i_ ± SEM (M). Agonist potency is expressed as apparent pEC_50_ ± SEM (M). Data are the mean of at least three independent experiments, each of them performed using at least six different ligand concentrations. * *p* < 0.05, ** *p* < 0.01, Paired *t*-test.

**Table 4 pharmaceutics-15-01845-t004:** Melatonin content in plant extracts measured using HPLC/MS.

Code	Scientific Name	Melatonin Content (mg/g of Extract)	Estimated Melatonin Concentration in ERK Assay (Figure 10) (nM)
**Ex6**	*Melissa officinalis* L.	0.03	0.20
**Ex8**	*Eleutherococcus senticosus*	nd	nd
**Ex19**	*Rhodiola rosea* L.	nd	nd
**Ex24**	*Valeriana officinallis* L.	0.06	0.38

nd, not detectable (<0.01 ng/mL).

## Data Availability

Data are contained within this article and Appendix A.

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
