# Peer review of "Pistacia vera Extract Potentiates the Effect of Melatonin on Human Melatonin MT1 and MT2 Receptors with Functional Selectivity"

_pharmaceutics, 2023, doi:10.3390/pharmaceutics15071845_

Round 1

Reviewer 1 Report

In the present manuscript, Dr. Jockers and co-workers examined plant extracts for their potential ability to modulate the binding of melatonin to melatonin receptors. The extensive set of screening assays reveals several plants (e.g. Melissa and Valeria species) to hold measurable amounts of melatonin while others (Eleutherococcus, rose) contain compounds capable of enhancing receptor activation in selected assays.  As highlighted in the Abstract the outstanding finding is that Pistacia vera extracts scored positive in each of the assays presented here suggesting a significant content of melatonin its dried fruits. As such, the data of the current manuscript may have impact on the public excitement about the alleged melatonin content in pistachio nuts and what it could do to your wellbeing.

As mentioned by the authors, the first published measurement of melatonin in pistachio kernels has raised concern as the reported content of melatonin could not be determined independently. Some of the concern is due to the excessive levels ascribed to pistachio which is about 1000-fold higher than that of any other plant material (200 µg melatonin/g dried fruit vs. 100 ng/g). Similarly, the current data suggest that melatonin accumulates to high levels in Pistachio fruits. The authors determined the melatonin content in an alcoholic extract of dried nuts using two independent assays (HPLC-MS and ELISA) and come up with values of 4-5 mg melatonin/g extract. The authors relate the amount of melatonin to the extracted matter. What is the value when the amount is related to the dried kernel mass?

In order to address possible concerns about adulteration of the material used authors are requested to re-assess the melatonin contennt in Pistachio material  obtained from an independent source.

Although it is disconcerting to apply forensic criteria to basic research given the previous controversy a reliable statement regarding the issue of melatonin content in pistachia appears highly desirable.

In a revised manuscript the extraction and chemical determination methods should to be documented in a separate Figure/Table.  

Additional comment

Lines 155 - 158: The statement made here somewhat contradicts the bar diagram of Figure 3F. The text states that the effect of Ex18 on MT2  was reversed by pertussis toxin pretreatment of MT2-receptor expressing cells while the Figure reveals data heterogeneity with overlapping cAMP values. The stateent in line 158 <Similar results were obtained for MT1 (not shown)> therefore needs to be clarified..

Figure 10: headings should read MT1 and MT2.

Reviewer 2 Report

The manuscript "Pistacia vera extract potentiates the effect of melatonin on human melatonin MT1 and MT2 receptors with functional selectivity" provides relevant information on different plants with possible effects on sleep disorders and anxiety. However, the authors should clarify the following points:

- It should be clarified in the introduction what was the criteria for the selection of these plant species.

- Provide the NMR spectra of the extracts of the 25 plant species in the Supplementary Material.

-What was the criteria for making only aqueous or aqueous and hydroalcoholic extracts in the species under study. This should be clarified, since the solvents used can extract certain compounds. And the comparison between plant species would be biased.

- Before carrying out any biological assay, the cytotoxicity of the extracts must be determined, in order to determine the safe concentrations for biological assays.

- The section on results and discussions should be further developed. The authors must justify which compounds present in the extracts of the species are responsible for the activities tested.

- Once the majority or main compounds have been identified, the authors must suggest the possible mechanisms of action of the extracts.

Authors should review the journal's platform. To determine what type of language is used. Since, the manuscript presents parts in British English and others in American English.

Reviewer 3 Report

The manuscript entitled ”Pistacia vera extract potentiates the effect of melatonin on human melatonin MT1 and MT2 receptors with functional selectivity” has been reviewed. The manuscript needs a major revision before publishing. 

The abstract is concise and the authors summarized all points including aim, methodology, results and conclusion. 

The introduction needs minor changes to provide a clear overview of the research. Justification of the study is missing. Few sentences need re-arrangement and rewriting to provide a better understanding of the topic to the audience. 

Methodology needs supporting references, (wherever missing, eg. Test of solubility 468, Cell culture of HEK293 cells, Determination of Melatonin content by ELISA) that have been published earlier, this will ensure the study is replicable and can be validated by other researchers. 

In the result section, the figure legend should follow a uniform font size. 

The discussion needs supportive recent references for the statements that are related to the current study. I could find a few 2020 and 2021, 2022 references.

The author should provide a clear and concise conclusion that summarizes the main findings of the study and their implications. No reference should be cited in conclusion.

A thorough checking is needed in terms of grammar, typing errors and usage of uniform font in figure (s) and abbreviations. 

By implementing these suggestions, the author can improve the overall quality of the manuscript.   

Moderate editing of English language required. Grammatical errors need considerable attention.

Round 2

Reviewer 1 Report

Thanks to the authors for clarification.

Reviewer 2 Report

The authors of the manuscript "Pistacia vera extract potentiates the effect of melatonin on human melatonin MT1 and MT2 receptors with functional selectivity" have made the respective corrections to the reviewers' observations. The manuscript in its current state can be considered for publication on the journal's platform.

Reviewer 3 Report

The manuscript needs a considerable revision in terms of scientific writing.

Minor corrections are needed. Corrections are needed throughout the manuscript. For example:

Line No. 138-Then, we then?? rewrite the sentence

Line No. 174 We next tested the effect..rewrite the sentence

Line No. 174 did not have any effect or "did not show any effect" rewrite the sentence

Line No. 199 Taken together, several extracts.. what does it mean?

Line No. 329 we asked the question whether the effect?? write in scientific way.

"We" is repeated many times in the manuscript that does not fit for a good scientific writting.

Extensive editing of English language required.
